# Resistance to Echinocandins Complicates a Case of *Candida albicans* Bloodstream Infection: A Case Report

**DOI:** 10.3390/jof7060405

**Published:** 2021-05-21

**Authors:** Laura Trovato, Dafne Bongiorno, Maddalena Calvo, Giuseppe Migliorisi, Albino Boraccino, Nicolò Musso, Salvatore Oliveri, Stefania Stefani, Guido Scalia

**Affiliations:** 1U.O.C. Laboratory Analysis Unit, A.O.U. Policlinico-San Marco, 95123 Catania, Italy; maddalenacalvo@gmail.com (M.C.); gpp.miglio@gmail.com (G.M.); lido@unict.it (G.S.); 2Department of Biomedical and Biotechnological Sciences, University of Catania, 95123 Catania, Italy; dbongio@unict.it (D.B.); nmusso@unict.it (N.M.); oliveri@unict.it (S.O.); stefanis@unict.it (S.S.); 3U.O.C. Anesthesia and Intensive Care, Ospedale Garibaldi-Nesima, Azienda di Rilievo Nazionale e Alta Specializzazione ‘Garibaldi’ Catania, 95122 Catania, Italy; albibor@gmail.com

**Keywords:** *Candida albicans*, bloodstream, echinocandins, resistance, mutations, FKS1

## Abstract

Invasive candidiasis is known to be one of the most common healthcare-associated complications and is caused by several *Candida* species. First-line drugs, particularly echinocandins, are effective, but there are increasing reports of resistance to these molecules, though rarely related to *C. albicans*. Even though the rate of echinocandins resistance remains low (<3%), sporadic cases are emerging. Here, we present a case of bloodstream infection by a pan-echinocandin-resistant *Candida albicans* affecting a critically ill patient, who died in an intensive care unit following therapeutic failure and multiple organ dysfunction syndrome. This case highlights the need to suspect pan-echinocandin resistance in patients with prolonged echinocandin exposure, particularly in the presence of urinary tract colonization. Our study shows the importance of sequencing to predict therapeutic failure in patients treated with echinocandins and persistent candidemia.

## 1. Background

*Candida* spp. normally colonizes healthy individuals asymptomatically, but it is also an opportunistic pathogen that can cause severe complications such as bloodstream infections [1]. The incidence of invasive candidiasis (IC) has increased significantly in recent years, with *Candida albicans* being the first cause of bloodstream infection followed by *Candida glabrata*. Patients on broad-spectrum antibiotics, immunosuppressed patients or patients with central venous access devices are more likely to develop fungal infections, including candidemia [2,3]. Antifungal therapy is a critical component of patient care, given the limited availability of antifungal drug classes. Echinocandins are recommended for the first-line treatment of invasive candidiasis, though in recent years *Candida* isolates with acquired resistance have been reported more frequently [3,4,5,6,7]. Echinocandins selectively inhibit glycosyltransferase 1,3-β-D-glucan synthase, the enzyme responsible for the biosynthesis of a fundamental structural component of the cell wall, the oligosaccharide 1,3-β-D-glucan. Today, three molecules are available with this function: caspofungin, micafungin and anidulafungin. Resistance to these molecules is predominantly associated with hotspot (HS) mutations in the 1,3-β-D-glucano synthase gene (FKS). Specifically for *C. albicans*, mutations responsible for echinocandins resistance were associated with two different hotspots in the FKS1 (or GSC1) portion [8]. However, with the exception of *C. glabrata*, resistance to echinocandins remains relatively low, being <3% for *Candida albicans* and most *Candida* spp. [9,10].

We report the case of a COVID 19-negative patient who developed bloodstream infection by a pan-echinocandin-resistant *Candida albicans* isolate. The patient died from multiple organ dysfunction syndrome before being able to benefit from a potentially effective antifungal therapy.

## 2. Case Report and Results

In May 2020, a 71-year-old man with a previous history of ischemic cardiac disease, stroke and hemiparesis was admitted to the Intensive Care Unit (ICU) of the Garibaldi Hospital in Catania for intensive monitoring after evacuation of a neck abscess. The patient was complaining of dysphagia and one-week fever treated with amoxicillin, corticosteroids and paracetamol. White blood cell count (WBC) was 28,700/mm^3^, C-reactive protein (CRP) 20.40 mg/dL, procalcitonin (PCT) 1.89 mcg/L, creatinin 1.4 mg/dL and lactate 3.4 mmol/L. Based on these findings, a septic event was suspected and empiric therapy with linezolid (1200 g/day) and piperacillin/tazobactam (18 g/day) was initiated. On the same day, a blood culture and bacteriological and mycological surveillance cultures (sputum and urinary samples, nasal and rectal swabs) were obtained. *C. albicans* was the only isolate detected in a nasal swab and in urinary and sputum samples. *C. albicans* was considered as a colonizer. A serum 1,3-β-D-glucan assay (Fungitell; Associates of Cape Cod Inc., Falmouth, MA, USA) was performed, revealing a positive result with a value of 114 pg/mL. In consideration of the patient’s medical history, a computed tomography (CT) of the neck and chest was performed on day 4. Bronchoalveolar lavage (BAL) and new surveillance cultures (urinary sample, nasal and rectal swabs) were obtained to investigate the possible presence of an infectious disease causing pneumomediastinum. *Candida albicans* was isolated from a nasal swab as well as urinary and a BAL samples.

On day 7, some inflammatory parameters were still high (WBC 24,800/mm^3^ and CRP 21.89 mg/dL), but a decrease in PCT (0.10 µg/L) was also observed and the patient was no longer febrile. He was then moved to the otolaryngology unit with the indication to continue empiric therapy with linezolid and piperacillin/tazobactam and to start parenteral nutrition. On day 8, due to worsening of general condition and a febrile episode, a new blood culture and several surveillance cultures (broncho-aspirate and urinary samples) were obtained. Parenteral nutrition and broad-spectrum treatment with linezolid (1200 mg/day) were continued, whereas piperacillin/tazobactam was switched to meropenem (3 g/day).

On day 9, the patient developed respiratory failure and was moved to the thoracic surgery unit due to pleural effusion in the right lung. On day 13, positive blood cultures for *Candida albicans* were reported; the same was observed for broncho-aspirate and urine. According to the guidelines for the management of candidemia and on the basis of in vitro susceptibility to antifungal agents, clinicians decided to start caspofungin at 50 mg daily (following a 70 mg loading dose on day 1), even if no initial sensitivity test was performed to support this choice [11]. On day 28, the patient developed respiratory failure and the onset of multiple organ dysfunction syndrome and was moved back to the ICU (Figure 1). On day 29, BAL and new surveillance cultures (urinary sample, nasal and rectal swabs) were obtained and a *Stenotrophomonas maltophilia* strain was isolated from both the nasal swab and the BAL sample, while *Candida albicans* was isolated from the nasal swab and urinary sample). No COVID-19 genome was detected in the BAL sample. A new antibiotic therapy with linezolid (1200 mg/day), trimetroprim/sulfametoxazole (16 g/day) and fosfomycin (16 g/day) was initiated, while antifungal treatment with caspofungin was continued. On day 33, serum levels of β-glucan were tested to monitor the course of the invasive infection, with a positive result of >523 pg/mL. On the same day, a new blood culture showed presence of *Candida albicans* (Isolate 2). On day 40, a new microbiological examination was performed on BAL, urinary and rectal samples: *S. maltophilia* was still isolated from the BAL sample, while the urinary sample revealed colonization by *C. albicans* and the rectal swab was positive for *Acinetobacter baumannii*.

Antibiotic therapy with colistin (9 million IU/day) was added to the previous treatment. On day 42, a new blood culture showed presence of *Candida albicans*. On day 44, based on in vitro antifungal susceptibility results, caspofungin was replaced by amphotericin B (400 mg/day). Unfortunately, the patient’s clinical conditions worsened before he could possibly benefit from the antifungal therapy change. On day 46, the patient died of multiple organ dysfunction syndrome. The clinical/microbiological monitoring is summarized in Figure 2. Table 1 summarizes the results of antifungal susceptibility testing and FKS1 gene sequencing all both *C. albicans* isolated. Identification of the species was confirmed by Matrix-assisted laser desorption ionization time of flight mass spectrometry (MALDI-TOT MS) on a Microflex LT (Bruker Daltonics, Bremen, Germany) platform. Isolates 1 and 2 were uniformly susceptible, while Isolate 3 was resistant to anidulafungin, micafungin and caspofungin, with MICs of 1, 4 and 8 µg/mL, respectively (SensititreYeastOne^®^ method; Thermo Fisher Scientific, Cleveland, OH, USA), according to the Clinical and Laboratory Standards (CLSI) clinical breakpoints [12]. Except for these three echinocandins, all isolates were susceptible to fluconazole (MIC, 0.25 mg/L), itraconazole (MIC, 0.06 mg/L), voriconazole (MIC, <0.008 mg/L), 5-fluorocytosine (MIC, <0.06 mg/L) and, according to the epidemiological cutoff values established by the CLSI [13], they also expressed wild-type susceptibility to amphotericin B (MIC, 0.5 mg/L) and posaconazole (MIC, 0.03 mg/L). The effectiveness of isavuconazole against all major *Candida* species has been evaluated [14]. According to these reports, the susceptibility test for isavuconazole was performed using the MIC strip method (Liofilchem, Roseto degli Abruzzi, TE, Italy) [15,16] and showed an MIC of 0.5 mg/L. However, the isolate could not be classified as either sensible or resistant to isavuconazole, according to the EUCAST breakpoint tables for the interpretation of MICs [17].

In consideration of the susceptibility results, molecular typing of the HS1 and HS2 of FKS1 genes was performed. For DNA extraction, the PathoNostics Extraction Kit was used following the manufacturer’s instruction. DNA was quantified using the Qubit^®^ 3.0 Fluorometer (Cat No. Q33216, life Technolo-gies, Thermo Fisher Scientific, Monza, Italy) and the fluorimeter Qubit dsDNA BR Assay Kit (Cod. 32850, Invitrogen, Thermo Fisher Scientific, Monza, Italy). DNA was used for molecular typing of the HS1 and HS2 of FKS1 gene, performed using the Sanger method. The HS1 region was amplified using primers previously published [18]. Primers used in HS2 region amplification were: C.aFSK1HS2F 3′-TGAGGATTGAAAATGAATGGGGA-5′ and C.aFSK1HS2R 3′-GCTTTAGAAACACCACCTCTAGT-5′; the amplicon size was 860 bp, Tm 60 °C. This set of primers was designed in house and analyzed with the online software: http://www.premierbiosoft.com/netprimer/ (accessed on 27 July 2020). Both PCR amplifications were carried out in a Veriti Thermal Cycler (Applied Biosystems, Ther-moFisher, Monza, Italy) in a total volume of 25 µL containing 2× Multiplex PCR Master Mix (cat. No. BR0200804, biotechrabbit GmbH, Hennigsdorf, Germany), 10 ng template DNA and verified agarose gel at 1.5% stained with SYBR Safe DNA Gel Stain (Cod. S33102, Invitrogen, Thermo Fisher Scientific, Monza, Italy). Obtained amplicons were purified, quantified and sequenced as previously described [19]. Obtained sequences were analyzed with BLAST (https://blast.ncbi.nlm.nih.gov/Blast.cgi (accessed on 7 October 2020)) and UNIPROT (https://www.uniprot.org/ (accessed on 7 October 2020)). Comparing our sequences to the reference strain (Ca22chr1A_*C_albicans*_SC 5314:505969.511662 (accessed on 7 October 2020)), we only found two homozygous single-nucleotide polymorphisms (SNP) in HS1, one of which is a silent mutation (A394G, reference strain position) that does not determine any alteration in the protein sequence. The other is a missense mutation (T398A, reference strain position) which was responsible for alteration in the 1,3-β-D-glucan protein sequence. In particular, the S645P mutation, as already described by Garnaud, results in transmembrane localization of the putative binding domain of the echinocandins, resulting in reduced affinity for these antifungals [8,20].

## 3. Discussion

We reported the case of a patient who developed bloodstream infection by a pan-echinocandin-resistant *Candida albicans* isolate acquired following echinocandin exposure. Our patient had several risk factors for the development of invasive candidiasis (extensive use of antibiotics, protracted hospitalization, *Candida* colonization index > 0.5). *Candida albicans* surface colonization of mucous membranes often represents the first stage of complication in a critical patient. The intensive care unit setting enables this yeast to invade the bloodstream, causing an increase in the mortality rate of hospitalized patients. Currently, the rate of echinocandin resistance remains low in this species and was mainly reported in isolates of *Candida glabrata* [21,22,23,24].

The main mechanism involved in echinocandin resistance of *Candida* spp. is related to mutations in hot-spot regions (HS1 and HS2) of the FKS1 gene, encoding the catalytic subunit of β-(1,3)-glucan synthase. The physiological role of the FKS2 and FKS3 genes in the onset of resistance among *Candida* species appears to be less important and is still widely debated [22]. Acquired mutations in the FKS1 and FKS2 genes have been predominantly found at position 645 (Serine), S645F (serine to phenylalanine), S645P (serine to proline), and S645Y (serine to tyrosine). Amino acid substitutions cause a change in the cell wall pattern, making it extremely difficult for echinocandins to hit their target. Mutations in the FKS genes are associated with high MIC values and low clinical response. These mutations usually result in pan-echinocandin resistance and affect mainly *C. glabrata* (prevalence range 2–13%) and, more rarely, *C. albicans* (<1%) [6,25]. In our case, a missense mutation S645P was observed in the HS1 region of the FKS1 gene of the *Candida albicans* isolate (Isolate 3), associated to echinocandin resistance. Echinocandin resistance can rapidly develop after initiation of treatment, but it can also appear after prolonged therapy, and caspofungin is associated with a higher risk of inducing FKS mutations in comparison to other echinocandins [23,26]. According to the literature data, we can say that echinocandin resistance is indeed more common in patients with long hospitalization and recurrent candidemia undergoing prolonged antifungal treatment [25,27]. In our case, the patient was put on caspofungin on day 13, as the first isolate of *C. albicans* susceptible to all antifungal agents tested was detected from blood cultures (Isolate 1). Following worsening of clinical conditions, respiratory failure and multiple organ dysfunction, the patient was moved back to the ICU and continued on caspofungin. On day 33, a new episode of candidemia was discovered: *C. albicans* was still detected (Isolate 2) and its susceptibility pattern was the same as in the first one. Only on day 42 did our patient develop an episode of candidemia with an echinocandin-resistant isolate of *C. albicans* (Isolate 3), following prolonged therapy with caspofungin (29 days). We demonstrated that Isolate 3, compared to Isolates 1 and 2 (wild type), had increased MICs for echinocandins, and all values were higher than the CLSI resistance breakpoints [12]. Besides prolonged antifungal treatment with echinocandins, another risk factor for the development of acquired FKS mutations is the potential existence of hidden reservoirs of *Candida* spp. (on abdominal cavity and mucosal surfaces), where there is no regular drug penetration [28,29]. Probably, in our case, the persistent urinary tract colonization also represented a risk factor for acquired FKS mutations and the appearance of echinocandins resistance during treatment further aggravated the adverse prognosis of candidemia. Despite early replacement of antifungal treatment, the patient died in two days due to the general worsening of his clinical conditions. This case further confirms the importance of quickly performing susceptibility tests upon isolation of *Candida* spp from sterile sites, especially when the infection appears not to respond to the antifungal treatment for a long time, as also stated in the current guidelines for the management of candidemia and invasive candidiasis of the Infectious Diseases Society of America (IDSA) [11]. Due to significant interlaboratory variability for caspofungin MICs, routine testing derived by CLSI or EUCAST methodology may lead to incorrect results especially when the isolate was detected to be resistant. Accordingly, molecular analysis is recommended for confirmation of resistance. Furthermore, SensititreYeastOne^®^ method should be routinely used to test in vitro susceptibility to caspofungin due to low MIC variability [30].

In conclusion, this case highlights that even though antifungal resistance in *C. albicans* is uncommon, individual isolates may not necessarily follow this general pattern and that susceptibility testing, as well as FKS sequencing, are indispensable to predict therapeutic failure in patients treated with echinocandins and persistent candidemia.

## Figures and Tables

**Figure 1 jof-07-00405-f001:**
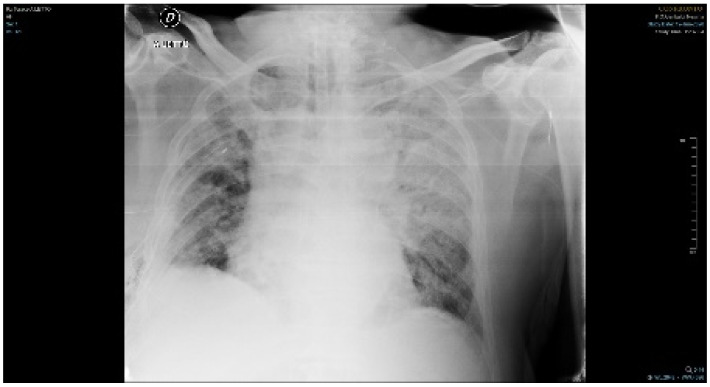
Chest radiograph shows several non-specific findings such as diffuse opacities, which correlate with clinical evidence of respiratory failure.

**Figure 2 jof-07-00405-f002:**
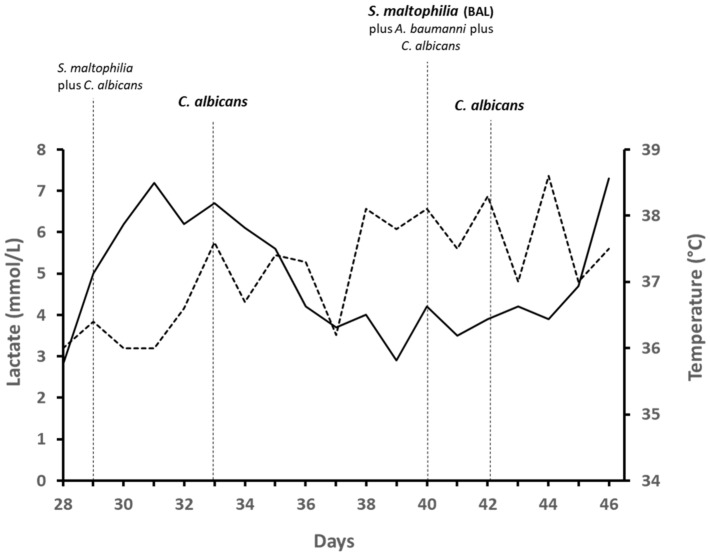
Clinical/microbiological monitoring during the patient’s stay at the ICU. Temperature (dashed line) and lactate (solid line) trends are shown. Isolates from blood culture (bold) and surveillance cultures (not bold) are shown.

**Table 1 jof-07-00405-t001:** Antifungal susceptibility testing and FKS1 gene sequencing for all *C. albicans* isolated.

Isolate	Day	MIC (mg/L) for Antifungal Agents									MIC Strip (mg/L)		FKS Hotspot Mutations
		AMB	AND	CAS	MFG	FCT	ITC	VRC	POS	FLZ	ISA	FKS1 HS1	FKS1 HS2
*C. albicans*	8	0.5	0.015	0.03	0.008	<0.06	0.06	<0.008	0.03	0.25	0.5	Wild type	Wild type
*C. albicans*	33	0.5	0.015	0.03	0.008	0.008	0.06	<0.008	0.03	0.25	0.5	Wild type	Wild type
*C. albicans*	42	0.5	1	8	4	0.008	0.06	<0.008	0.03	0.25	0.5	S645P	-

MIC, minimum inhibitory concentration; AMB, amphotericin; AND, anidulafungin; CAS, caspofungin; MFG, micafungin; FCT, flucytosine; ITC, itraconazole; VRC, voriconazole, POS, Posaconazole; FLZ, fluconazole; ISA, isavuconazole. CLSI breakpoints [mg/L]: anidulafungin, caspofungin and micafungin S ≤ 0.25, R ≥ 1.

## Data Availability

All data generated or analyzed during this study are included in this published article.

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
