# Peer review of "Resistance to Echinocandins Complicates a Case of Candida albicans Bloodstream Infection: A Case Report"

_jof, 2021, doi:10.3390/jof7060405_

Round 1
Reviewer 1 Report
The case report describes a patient who developed bloodstream infection by a pan-echinocandin resistant Candida albicans after prolonged administration of caspofungin emphasizing the importance of susceptibility testing and the FKS sequencing of individual isolates of Candida spp. to predict therapeutic failure in patients treated with echinocandins, especially when the infection appears not to respond to the antifungal treatment for a long time.
In my opinion the susceptibility testing could be sufficient to have this information, considering that the FKS sequencing requires highly specialized equipment and personnel, and that the resistance is demonstrated by MIC increase for all echinocandins tested.
I suggest authors specify the MIC strip method used for isavuconazole susceptibility (lines 145-146).
Author Response
According to revisor’s suggestion, the MIC strip method for isavuconazole has been specified in the line 147.
Reviewer 2 Report
The article in question is well written , has a clear and objective language. The theme presented Resistance to echinocandins complicates a case of Candida albicans bloodstream infection: a case report, is very relevant and up to date. The study has as its central theme an assessment of the resistence of microorganisms to antifungal agents that has been growing recently. In tis case, an improved study was made of some antifungals against C. albicans isolated fromcandidiasis invasive. The study has demonstraded several factors that should be considered regarding the behavior of fungal species in relation to antifungal agents used in the treatment of infections caused by fungi. In particular, the question of the resistence mechanisms acquired by this group of microorganisms. The various stages of study are well determined, well written and excellent understanding. The figures are consistent with the results. Bibliography is well to the topic studied. It is necessary to correct the writing of C. albicans: move to italics- part of the discussion.

Author Response
The writing of C. albicans has been moved to italics in all the lines of the discussion.